# Bile proteome reveals biliary regeneration during normothermic preservation of human donor livers

Adam M. Thorne [1,2], Justina C. Wolters [3], Bianca Lascaris[1,2],
Silke B. Bodewes [1,2], Veerle A. Lantinga [1,2], Otto B. van Leeuwen [1,2],
Iris E. M. de Jong[1,2], Kirill Ustyantsev[4], Eugene Berezikov [4], Ton Lisman[1,5],
Folkert Kuipers [3,4], Robert J. Porte[1,6,7] & Vincent E. de Meijer [1,2,7] ✉

Normothermic machine perfusion (NMP) after static cold storage is increasingly used for preservation and assessment of human donor livers prior to transplantation. Biliary viability assessment during NMP reduces the risk of post-transplant biliary complications. However, understanding of molecular changes in the biliary system during NMP remains incomplete. We performed an in-depth, unbiased proteomics analysis of bile collected during sequential hypothermic machine perfusion, rewarming and NMP of 55 human donor livers. Longitudinal analysis during NMP reveals proteins reflective of cellular damage at early stages, followed by upregulation of secretory and immune response processes. Livers with bile chemistry acceptable for transplantation reveal protein patterns implicated in regenerative processes, including cellular proliferation, compared to livers with inadequate bile chemistry. These findings are reinforced by detection of regenerative gene transcripts in liver tissue before machine perfusion. Our comprehensive bile proteomics and liver transcriptomics data sets provide the potential to further evaluate molecular mechanisms during NMP and refine viability assessment criteria.

Liver transplantation is the only curative treatment option for patients with end-stage liver disease. However, a persistent disparity remains between supply and demand for donor organs suitable for transplantation, resulting in a waiting list mortality of up to 20%[1]. In turn, this has resulted in an increasing use of high-risk grafts from suboptimal, extended criteria donors (ECD). These ECD grafts include livers from donation after circulatory death (DCD) donors, and donors with significant comorbidities such as advanced age, diabetes mellitus and livers with high fat content[2].

Static cold storage has traditionally been used for preservation of donor livers prior to transplantation. While cold storage remains sufficient for livers considered optimal for transplantation, ECD grafts are more vulnerable to ischemia-reperfusion injury-related complications post-transplant. The most notable of these complications are primary non-function and ischemic cholangiopathies, including non-anastomotic strictures, which remain the 'Achilles heel' of ECD liver transplantation[3,4].

As an alternative to static cold storage, ex-situ normothermic machine perfusion (NMP) has become an increasingly popular method

[1]Department of Liver Transplantation and HPB Surgery, University of Groningen and University Medical Center Groningen, Groningen, the Netherlands.
[2]UMCG Comprehensive Transplant Center, Groningen, the Netherlands. [3]Department of Pediatrics, University of Groningen, University of Groningen and University Medical Center Groningen, Groningen, the Netherlands. [4]European Research Institute for the Biology of Ageing (ERIBA), University of Groningen and University Medical Center Groningen, Groningen, the Netherlands. [5]Surgical Research Laboratory, Department of Surgery, University of Groningen, University Medical Center Groningen, Groningen, the Netherlands. [6]Present address: Erasmus MC Transplant Institute, Department of Surgery, Division of HPB and Transplant Surgery, University Medical Center Rotterdam, Rotterdam, the Netherlands. [7]These authors jointly supervised this work: Robert J. Porte, Vincent E. de Meijer. ✉e-mail: v.e.de.meijer@umcg.nl

for preservation, transportation, and assessment of ECD livers prior to transplantation[5,6]. During NMP, the donor liver is maintained at 35–37 °C and metabolism is restored. This allows for functional assessment of the graft and provides the opportunity to assess key processes that can be used to determine organ viability[7–9]. These processes can be separated into two distinct areas: first, hepatocellular viability, focusing on the metabolic functionality of the liver parenchyma, is assessed by criteria such as lactate clearance and bile production. Second, biliary viability that focuses on the functional capacity of cholangiocytes, which is assessed by specific biochemical composition of the bile[10–12]. Current viability assessment strategies vary between centers, and some clinical studies only account for hepatocellular viability criteria, which maintain a high incidence of post-transplant cholangiopathies associated with DCD liver transplantation[7,13]. The addition of biliary viability criteria, such as bile pH, glucose reabsorption and bicarbonate secretion, for selection of viable liver grafts has been explored by us and others[8,14]. The use of both hepatocellular and biliary viability criteria in selection of suboptimal livers from ECD donors has led to the successful transplantation of these grafts with minimal incidence of post-transplant cholangiopathies[8,9,14,15].

However, the mechanisms of biliary injury and recovery thereof, following static cold storage, remain poorly understood. Recently, De Jong et al. found that preserved biliary microvasculature and peribiliary glands (PBGs) correlate with bile chemistry and capacity for regeneration[16]. Insight into these mechanisms may further improve preservation and selection of ECD livers. Therefore, we performed an in-depth proteomics analysis of bile samples collected from human ECD livers during NMP. We aimed to identify molecular pathways linked to biliary injury, function, and potential regeneration over the course of the perfusion, and to investigate whether biliary protein profiles can assist in identifying livers considered acceptable for transplantation.

This study demonstrates substantial changes in the biliary proteome during NMP. We found an increased abundance of proteins related to biliary regeneration in livers that had acceptable bile chemistry for transplantation. These differences were also reflected by transcripts from liver tissue taken before machine perfusion. Our findings shed a light on mechanisms associated with the recovery of the biliary tree following ischemia-reperfusion injury and provide a novel, comprehensive resource of potential targets for future therapeutic interventions to mitigate biliary injury during liver NMP.

## Results

### Study cohort
All consecutive liver grafts undergoing viability assessment using ex-situ NMP between 30th March 2019–30th June 2022 were eligible for this study, totaling 55 individual donor livers. Donor characteristics for the livers included in this analysis are provided in Table 1 (individual data are displayed in Table S1). All 55 donor livers were from DCD type III or V donors. The median donor age was 66 years (IQR; 61–69 years). Median cold ischemia time was 257 min (IQR; 223–288 min) and the median donor risk index (DRI) was 2.878 (IQR; 2.553–3.211). All livers were assessed according to predefined, previously established viability criteria ('traffic light' system), based on both hepatocellular and biliary chemistry (Table S2)[8]. Of the 55 livers, 35 (70%) were considered viable based on a combination of hepatocellular and biliary viability criteria and were subsequently transplanted, while 20 (30%) did not meet the viability criteria, after which the perfusion was terminated, and the organ discarded (Table S3). One liver was declined solely on hepatocellular criteria (number 26, Table S3), based on inadequate lactate clearance, whereas 19 livers were declined based on a combination of hepatocellular and biliary viability criteria. For cholangiocellular assessment we used a 'biliary viability score' to quantify how well each liver met the biliary viability criteria. Based on bile biochemical

## Table 1 | Donor characteristics for included livers

| | All (n = 55) | |
|---|---|---|
| Donor characteristics | | |
| Age (years) | 66 | (61–69) |
| Body mass index (kg/m²) | 26 | (24–28) |
| Gender | | |
| Male | 38 | (69%) |
| Female | 17 | (31%) |
| Cause of death | | |
| Trauma | 10 | (18%) |
| CVA | 22 | (40%) |
| Anoxia | 15 | (27%) |
| Other | 8 | (15%) |
| Donor type | | |
| DCD | 55 | (100%) |
| DBD | 0 | (0%) |
| Time from withdrawal of life support to circulatory arrest (min) | 16 | (14–26) |
| Time from circulatory arrest to cold perfusion (min) | 16 | (15–18) |
| Functional donor warm ischemia time* (min) | 31 | (26–34) |
| Last sodium (mmol/L) | 143 | (139–146) |
| Last AST (U/L) | 37 | (27–83) |
| Last ALT (U/L) | 30 | (18–62) |
| Last GGT (U/L) | 36 | (20–95) |
| Last ALP (U/L) | 69 | (57–89) |
| Hepatectomy time (min) | 39 | (33–41) |
| Static cold ischemia time (min) | 257 | (223–288) |
| DRI# | 2.878 | (2.553–3.211) |

Continuous data are presented as median (IQR), categorical data as a number (percentage).
#Validated scoring tool to assess the risk of liver graft failure. * Time from donor saturation <80% or mean arterial pressure <60 mm/Hg to initiation of in-situ cold flushing in the donor. Static cold ischemia time was defined as the time between initiation of cold flushing in the donor and start of DHOPE.
ALP alkaline phosphatase, ALT alanine aminotransferase, AST aspartate aminotransferase, CVA cerebral vascular accident, DBD donation after brain death, DCD donation after circulatory death, DRI donor risk index, GGT gamma glutamyl transferase.

composition, green values were assigned 2 points, orange values 1 point, and red values 0 points. The median biliary viability score for livers that were accepted or not accepted for transplantation was 8 and 1, respectively.

The total number of samples was n = 55 at 30 min and n = 54 at 150 min. We did not detect any protein in one bile sample at 150 min using mass spectrometry analysis. Of the 35 livers that were transplanted, 33 had bile samples available at the end of perfusion (End). A total of 142 bile samples were used for proteomic analysis (Fig. 1a).

### Recipient characteristics and outcomes
Recipient characteristics are provided in Table 2. Median model for end stage liver disease (MELD) score was 11 (IQR; 9–16). Median recipient age was 61 years (IQR; 53–66). Patient and death-censored graft survival 97% and 94%, respectively. There were two cases of re-transplantation due to chronic rejection and hepatic artery thrombosis and one patient death due to interstitial lung disease. There was one case of post-transplant ischemic cholangiopathy.

### Bile duct histology at baseline
To assess the severity of histological bile duct injury after static cold storage, extrahepatic bile duct biopsies obtained prior to machine

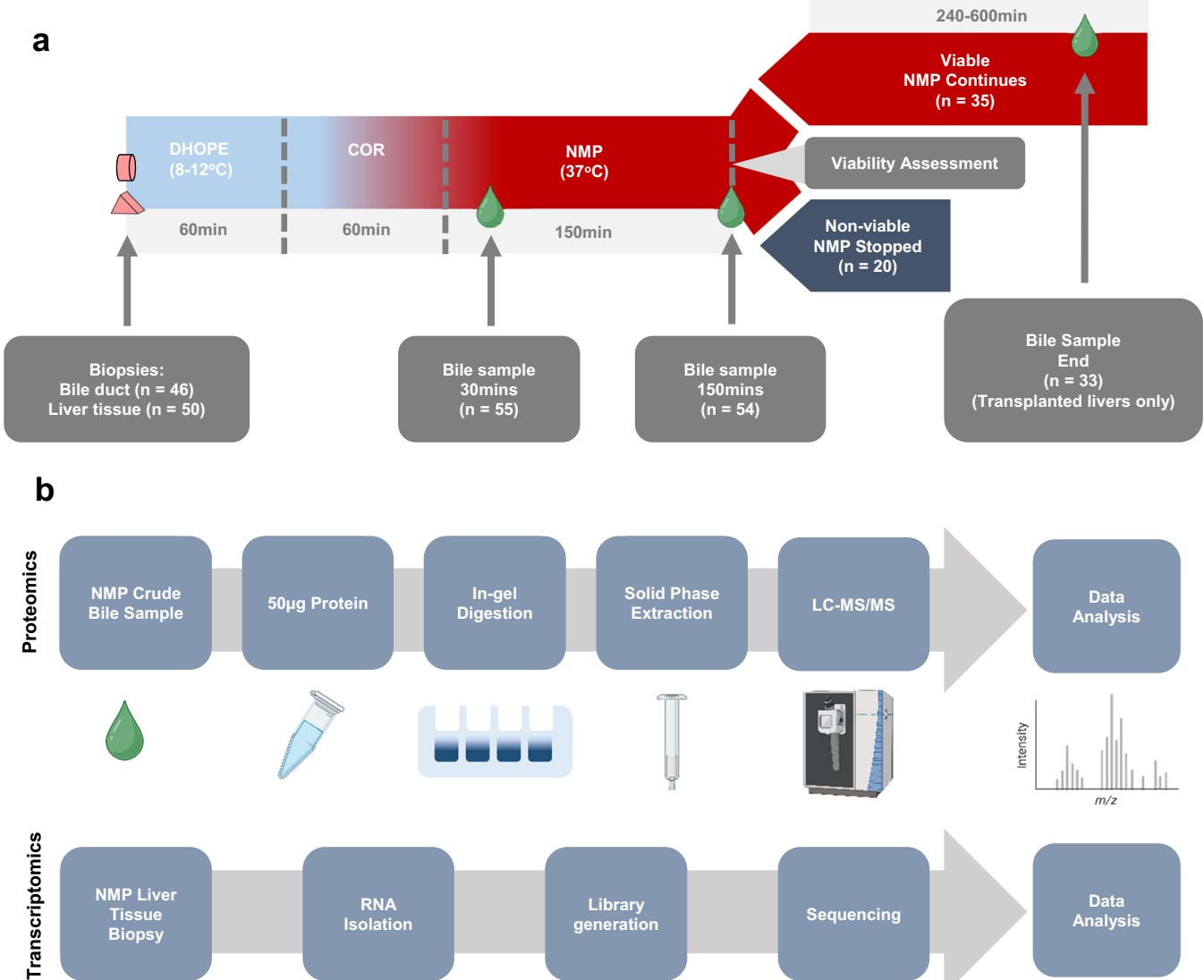

**Fig. 1 | Overview of machine perfusion and proteomics and transcriptomics workflows. a** Workflow schematic of the DHOPE-COR-NMP procedure. Sample numbers ($n =$ ) are displayed for each time point: 30 min after start of NMP (37 °C) and 150 min (2.5 h) NMP, the time viability assessment of the liver was conducted. For livers that were deemed viable and subsequently transplanted, bile samples were taken from a common time point towards the end of the perfusion (End), allowing a window of 120 min. Livers considered non-viable livers were not transplanted and the perfusion terminated at the time of viability assessment (150 min). **b** Workflow schematic of sample preparation for proteomics analysis by LC-MS/MS. Fifty micrograms of protein were taken from each crude bile sample and ran on an SDS-PAGE gel. Proteomics were reduced, alkylated, digested with trypsin and cleaned-up using solid phase extraction. Peptides were separated using Ultra High Performance Liquid Chromatography prior to analysis by MS/MS. RNA sequencing libraries were constructed according to the Smart-3SEQ protocol. Created with Biorender.com.

perfusion were retrospectively evaluated for injury using four well-established characteristics: vascular lesions, stromal necrosis, injury to periluminal peribiliary glands (PBG), and injury to deep PBGs[16,17]. A total of 46 bile duct biopsies were available from the 55 NMP livers. Livers with acceptable bile chemistry showed a trend towards lower levels of histological injury in all categories, although this was only significant for injury to deep PBGs ($p = 0.022$, Fig. 2a). Examples of bile ducts with low and high histological injury are depicted in Fig. 2b, c respectively.

When combined into a total histological bile duct injury (BDI) score, the scores ranged from 2 (least injured) to 14 (most injured), with a median score of 7. We used this median value to divide livers into groups of low and high total histological BDI score (Fig. 2d). Total BDI score was significantly different between livers with high biliary viability and those with low biliary viability, as based on biochemical biliary composition ($p = 0.046$, Fig. 2d).

Livers with low histological BDI scores (well-preserved bile ducts) were more likely to have high biochemical biliary viability ($n = 18$ [78%] vs $n = 5$ [22%]). However, a high histological BDI score (severely damaged bile ducts) did not discriminate between acceptable or inadequate biochemical biliary viability ($n = 11$ [52%] vs $n = 10$ [48%]; Fig. 2e). Thus, within the group livers with severely damaged bile ducts, still a large proportion (52%) could successfully be transplanted after NMP (11 out of 21). This suggests that either (1) histological examination of biopsies from the end of the common bile duct may not fully capture the health of the entire biliary tree; or (2) that an alternative mechanism—other than or in addition to initial histological injury—contributes to a favorable biochemical bile composition during NMP.

## Proteomics
To identify time-dependent changes in protein abundance patterns of bile produced during machine perfusion and potential differences

**Table 2 | Recipient characteristics**

| | Transplanted livers | |
|---|---|---|
| | (*n* = 35) | |
| Recipient characteristics | | |
| Recipient age | 61 | (53–66) |
| Recipient gender | | |
| Male | 15 | (56%) |
| Female | 20 | (44%) |
| Recipient body mass index (kg/m²) | 28 | (24–30) |
| Recipient MELD | 11 | (9–16) |
| Post-transplant outcomes | | |
| Actuarial graft survival | | |
| 3-months | 35 | (100%) |
| 6-months | 35 | (100%) |
| 12-months | 33 | (94%) |
| Retransplantation | 2 | (6%) |
| Actuarial patient survival | | |
| 12-months | 34 | (97%) |
| Peak ALT (U/L) | 606 | (367–1001) |
| Peak AST (U/L) | 988 | (573–1494) |
| Bilirubin day 7 (umol/L) | 12 | (8–22) |
| INR day 7 | 1.1 | (1.0–1.1) |
| GGT (IU/L) | | |
| day 30 | 86 | (57–148) |
| day 90 | 61 | (926–119) |
| day 180 | 53 | (21–114) |
| Alkaline phosphatase (U/L) | | |
| day 30 | 148 | (102–193) |
| day 90 | 138 | (95–195) |
| day 180 | 125 | (86–191) |
| Biliary complications | | |
| Non-anastomotic strictures | 1 | (3%) |
| Anastomotic strictures | 8 | (23%) |
| Bile leakage | 2 | (6%) |
| Primary non-function | 0 | (0%) |
| Hepatic artery thrombosis | 1 | (3%) |
| Acute rejection | 0 | (0%) |
| Chronic rejection | 1 | (3%) |
| Relaparotomy for: | | |
| Bleeding | 2 | (6%) |
| Bile leakage | 2 | (6%) |
| Other | 3 | (9%) |

Continuous data are presented as median (IQR), categorical data as a number (percentage).
*ALT* alanine aminotransferase, *AST* aspartate aminotransferase, *GGT* gamma glutamyl transferase, *ICU* intensive care unit, *INR* international normalized ratio, *MELD* model for end stage liver disease.

between livers with high or low biochemical biliary viability, we used an untargeted, data-independent acquisition mass spectrometry approach. With this strategy, a total of 4408 unique proteins were identified across all 142 samples. After preliminary analysis, we applied a valid value filter at each timepoint to reduce noise and remove less robustly identifiable proteins. The filter cut-off was defined using the median percentage presence of proteins in all samples (41.5%). From this, proteins present in at least 40% of the samples at each time point were kept, resulting in a total of 2865 proteins to be used for subsequent analysis (Fig. S1).

## Initial flush-out of cellular debris is followed by cellular signaling and secretory processes

To explore protein differences in bile over the course of machine perfusion, we compared all samples, across all three selected time-points: 30 min, 150 min and End. A heatmap using unsupervised hierarchical clustering for Euclidian distance of all proteins was performed at each time point. Using k-means clustering at 30 min, we identified four distinctive protein clusters across all time points (Fig. 3a). To reduce the complexity of the heat map, 30 min vs 150 min and 150 min vs End were compared using principal component analysis (PCA). Using PCA, we observed modest clustering of protein composition between samples taken after 30 min or 150 min (Fig. 3b). This separation was less apparent when comparing samples taken at 150 min and at the end of perfusion (Fig. 3c), indicating a more similar, but still changing, protein composition at later time points.

To assess changes in individual proteins, volcano plots were used to identify proteins that had a significant ($p < 0.05$) change in abundance of more than 2-fold. A total of 1296 proteins showed a significant >2-fold change when comparing samples taken at 30 min and samples taken at 150 min (Fig. 3d). Of these 1296 proteins, 1040 were enriched at 30 min, compared to 150 min. Gene ontology (GO) analysis with functional enrichment revealed that the significantly enriched proteins at 30 min included cellular components among which cell organelles, cytoplasmic and cell membrane proteins (associated with extracellular vesicles) were present; reflective of cellular debris. Biological process analysis at the same time point revealed significant enrichment of proteins involved in cellular metabolism, translation initiation and cellular localization. Considerably fewer proteins were enriched at 150 min when compared to 30 min, with just 256 significantly upregulated. These upregulated proteins were associated with immune response and cell secretion pathways (Fig. 3d).

Comparison of the protein pattern of samples taken after 150 min and at the end of perfusion revealed a total of 838 proteins that were significantly changed between the two groups, 255 of which were upregulated at the End time point. Functional enrichment analysis showed a continuing increase in proteins associated with secretion and cellular signaling, with a reduction of cellular debris over time (Fig. 3e).

Taken together, these findings demonstrate a flush-out of cellular debris and cellular activation during the first 30 min of NMP. This was followed by an increase in metabolic, signaling, and immune response processes at later time points. These changes illustrate a substantial shift in protein profile during machine perfusion, and a reduction of cellular damage after the first 30 min of NMP.

## Livers with high biliary viability showed early signs of cellular regeneration

In addition to the overall changes in the bile proteome during perfusion, we compared protein changes in bile between livers with high and low biliary viability scores. Using the same initial approach as with the longitudinal analysis, we first created a heat map to visualize global proteomic differences at 150 min (when liver viability is assessed; Fig. 4a). In addition, we separated the data into livers with low and high total histological BDI scores. Within the total BDI score groups, livers were further stratified into those with a high biliary viability score and low biliary viability score, based on biochemical bile composition. We observed mild clustering between biochemical biliary viability score groups in total histological BDI score classifications; however, it was not definitive. Even though five defined protein groups were defined using k-means, clear differences in protein abundance between groups (low and high BDI, as well as low and high biliary viability) were difficult to visually distinguish on a global proteome scale. As livers with low histological BDI have a high likelihood of being acceptable for transplantation (78%), we further investigated whether the composition of

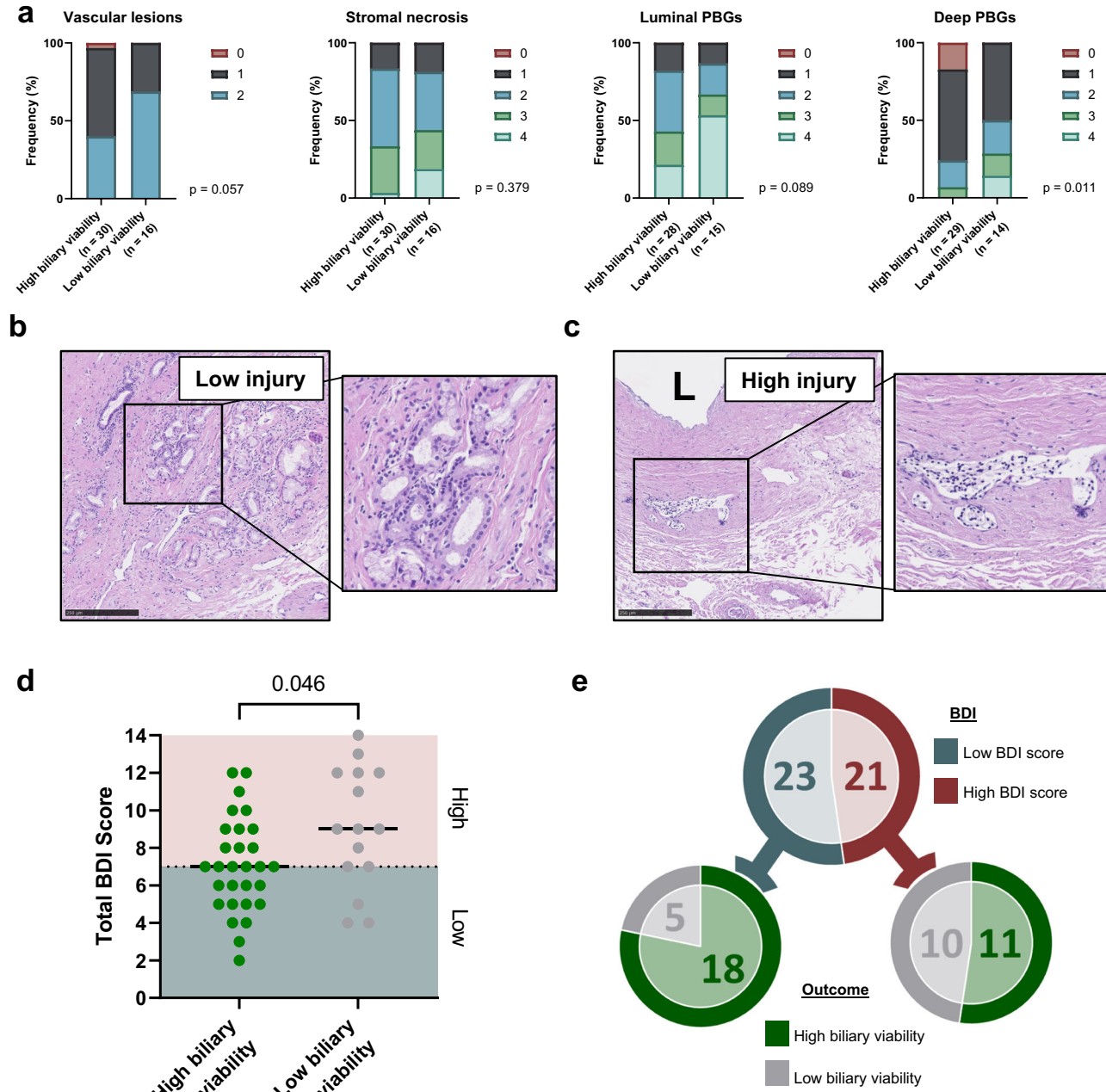

**Fig. 2 | Histological analysis of bile duct injury (BDI) prior to machine perfusion. a** Bar charts showing frequency (%) of injury score in livers with high vs low biliary viability score in four areas: vascular lesions, stroma necrosis, luminal peribiliary glands (PBGs) and deep PBGS. A hematoxylin and eosin-stained histological example of a low injury and high injury bile duct are shown in (**b**, **c**), respectively. Histological analysis was performed on $n = 46$ bile duct biopsies, however two of these were not possible to grade fully. Insets show an intact PBG cluster, located deep in the bile duct wall (**b**) and a damaged luminal PBG (**c**). L, lumen; scale bars 250 μm. **d** Dot-plot showing the total BDI score for livers with high and low biliary viability scores, as defined by the traffic light scoring system; 2 point for green values, 1 for orange, 0 for red. High and low histological BDI is depicted by the red (high injury, $n = 29$) and blue (low injury, $n = 15$) sections of the graph, based on the median histological BDI score (7) from all livers. $P$ values were calculated using two-tailed Mann–Whitney test. **e** Chart depicting distribution of livers in low and high BDI groups, and the subsequent biliary viability score of livers within each group.

the bile proteome from livers with high histological bile duct injury attributed to high or low biliary viability. The PCA of livers with high BDI at 150 min showed moderate separation between high and low biliary viability scores, reflecting the clustering observed in the heat map (Fig. 4b).

At 150 min of NMP, a total of 680 proteins were significantly different between livers with high vs low biliary viability. One-hundred and seventy-four of these were enriched in livers with a high biliary viability score, and 506 in livers with a low biliary viability score. GO analysis with functional enrichment of these proteins revealed a

significant upregulation of regenerative processes, such as cell population proliferation, cellular signaling, cellular migration, and immune response processes in livers with high biliary viability scores. Furthermore, we observed a marked increase in proteins related to extracellular exosomes. Livers with low biliary viability scores showed increased abundances of proteins involved in translation initiation, and various (small molecule) metabolic processes (Fig. 4c). These differences are also observed, to different degrees, in all other comparisons of histological BDI score and time point (BDI low; 30 min, BDI high; 30 min, BDI low; 150 min; Fig. S2). Observations of regenerative

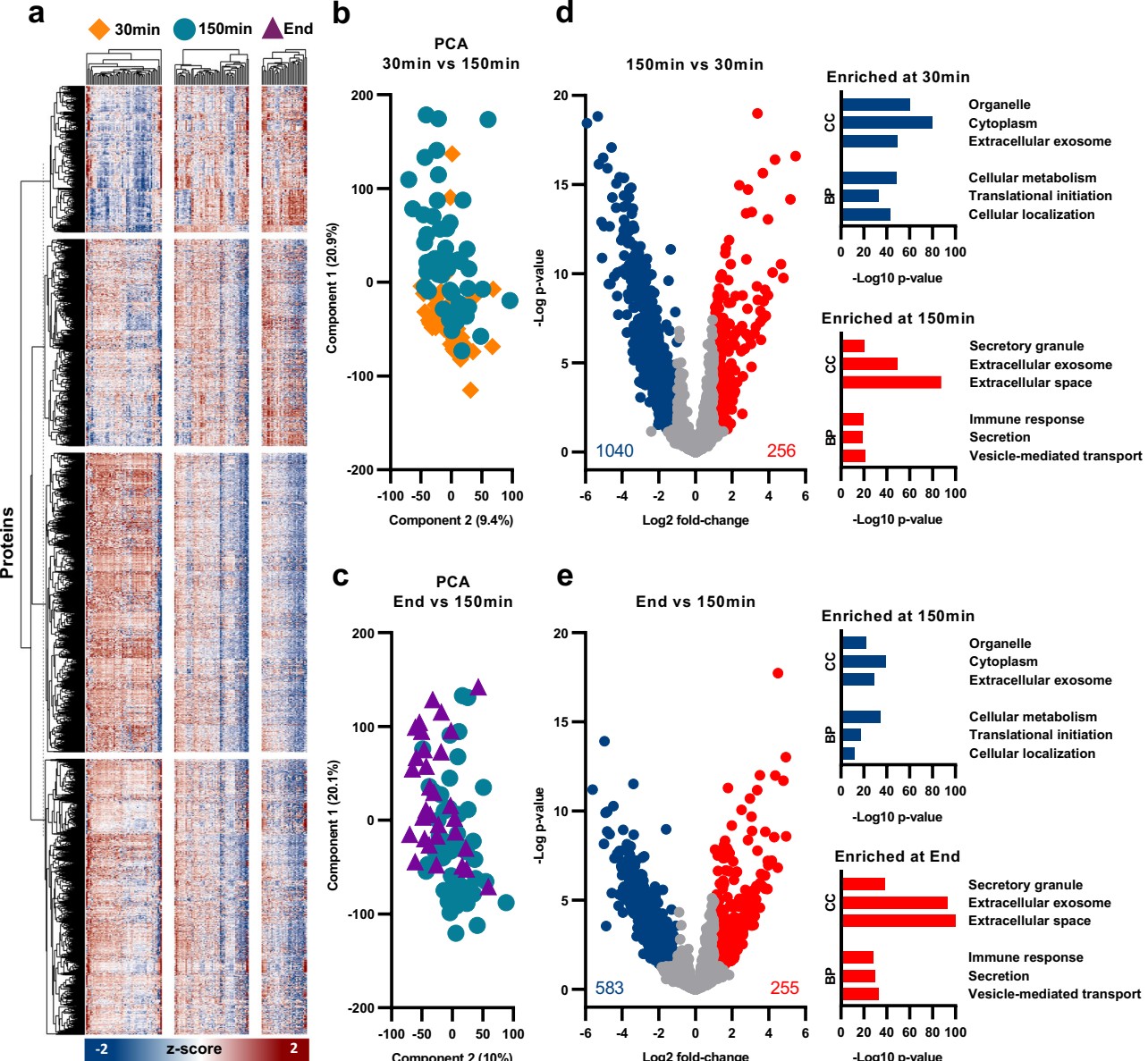

**Fig. 3 | Longitudinal protein abundance in all samples at 30 min, 150 min and End of NMP. a** Heat map showing z-score intensity of all proteins identified following 40% valid value filtering. Columns represent individual samples and are separated into three time points: 30 min, 150 min and End. Rows represent individual proteins and were separated using k-means clustering. Columns are clustered hierarchically. **b** Principal component analysis (PCA) comparing 30 min (orange diamond) and 150 min (blue circle). **c** Principal component analysis comparing 150 min (blue circle) and End (purple triangle). **d** Volcano plot showing significance and fold change of protein intensity at 150 min vs 30 min. Significant proteins ($p < 0.05$, >2-fold change) are highlighted red for upregulated (enriched at 150 min) and blue for downregulated (enriched at 30 min). Statistics were performed using a two-tailed Students $t$ test with a permutation-based FDR of 0.05 to assess multiple comparisons. Cellular component (CC) and biological process (BP) gene ontology (GO) pathways are displayed as −Log10 $p$ value. **e** Volcano plot showing significance and fold change of protein intensity at End vs 150 min. Significant proteins ($p < 0.05$, >2-fold change) are highlighted red for upregulated and blue for downregulated. Statistics were performed using a two-tailed Students $t$ test with a permutation-based FDR of 0.05 to assess multiple comparisons. The CC and BP gene ontology pathways are displayed as -Log10 $p$ value.

processes were not apparent in comparisons of histological BDI group only (high vs low BDI score; Fig. S3), suggesting that regenerative processes are not a result of the level of biliary injury the liver has sustained, but rather the ability of the liver to respond to that injury. This was further confirmed by comparison of livers with high and low biochemical biliary viability scores, regardless of histological BDI injury. Livers with high biochemical biliary viability show increased abundance of regenerative processes, albeit to a lesser degree, while livers with low biochemical biliary viability maintain abundance of translation and metabolic processes (Fig. S4). Furthermore, the abundance of individual proteins associated with these regenerative

processes showed a clear distinction between livers with high and low biliary biochemical viability scores in both high histological BDI (Fig. 4d) and low histological BDI (Fig. 4e).

**Mucins were strongly correlated with biliary viability and are detectable at the transcript level prior to machine perfusion**

After identifying significant differences in biliary proteins between livers with high and low biochemical biliary viability scores, we assessed the abundance of individual proteins for potential clinical application to aid the decision-making process. We analyzed all 2865 identified proteins for their capacity to differentiate between livers

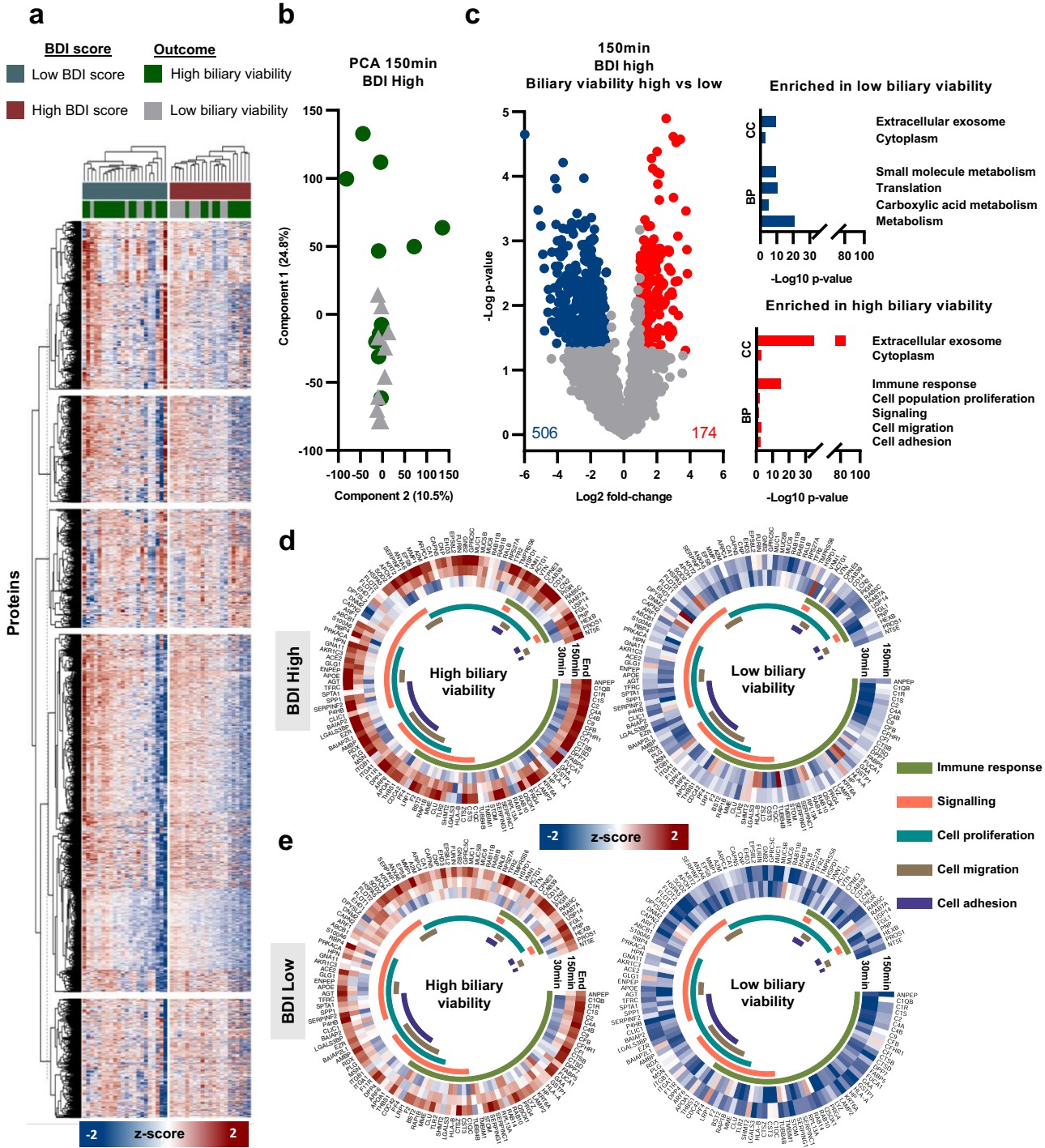

**Fig. 4 | Protein abundance between livers with high and low bile duct injury (BDI), and high and low biliary viability score. a** Heat map showing z-score intensity of all proteins identified following 40% valid value filtering. Columns represent individual samples at 150 min normothermic machine perfusion (NMP) and are split into two groups: low BDI and high BDI. Columns are further stratified into high (green) and low (gray) biliary viability score categories. Rows represent individual proteins. Columns and rows are clustered hierarchically. **b** Principal component analysis (PCA) of individual livers with high BDI at 150 min. Livers with high biliary viability scores are represented by green circles, low biliary viability scores with gray triangles. **c** Volcano plot showing significance and fold-change of protein intensity, comparing livers with high and low biliary viability scores within the high BDI group at 150 min. Significant proteins ($p < 0.05$, >2-fold change) are highlighted red for upregulated (enriched in high biliary viability score livers) and blue for downregulated (enriched in low biliary viability score livers). Statistics were performed using a two-tailed Students $t$ test with a permutation-based FDR of 0.05 to assess multiple comparisons. Cellular component (CC) and biological process (BP) gene ontology (GO) pathways are displayed as −Log10 $p$ value. Average (mean) z-score intensity of proteins involved in the five selected pathways (immune response, signaling, cell proliferation, cell migration and cell adhesion) in high and low biliary viability score livers within the high BDI group (**d**) and low BDI group (**e**).

with high and low biochemical biliary viability scores using receiver operator characteristic (ROC) area under the curve (AUC). Proteins were ordered based on their average AUC over comparisons within histological BDI group and time point (low BDI: 30 min, 150 min; high BDI: 30 min, 150 min). The top-30 highest average AUC (range 0.90–0.76) proteins are presented in Table S4. The top 3 proteins were FCGBP, MUC5B and MUC1, with average AUCs of 0.90, 0.86, and 0.85, respectively (Table S4).

To assess the possibility of identifying these proteins at the earliest time point possible, we took RNA sequencing data from parenchymal biopsies taken from the liver on the back table upon arrival at our center, prior to initiation of machine perfusion (Fig. 1a). Of the 50 biopsies available at baseline, 35 had sufficient RNA quality for RNA sequencing (RIN > 7). Normalized raw count data of transcripts from each of the top-30 identified proteins, categorized into livers with high and low biochemical biliary viability scores at 150 min NMP, are presented in Fig. S5 (all livers, regardless of BDI score), Fig. S6 (high BDI livers) and Fig. S7 (low BDI livers). Across all livers, regardless of severity of histological BDI, MUC5B, MUC1, and LCN2 presented as the only genes to show significant differences in expression between biliary viability groups ($p = 0.024$, $p = 0.021$, $p = 0.018$, respectively. Figs. 5c, e, S5). In high BDI livers, *MUC1* presented as the only gene with a significant difference between biliary viability groups ($p = 0.04$, Fig. S6). Other genes showed trends towards upregulation in livers with high biochemical biliary viability score (*MUC5B* [$p = 0.076$], *GPRC5C* [$p = 0.097$], *DPP4* [$p = 0.097$] and *NT5E* [$p = 0.097$], among others) but were not statistically significantly different (Fig. S6). These differences were less apparent in low histological BDI livers, suggesting the extent of liver injury during the donation and transportation process may be a driving factor for transcription initiation (and later translation to protein) of these genes (Fig. S7).

RNA sequencing raw count data for genes of the top-3 identified AUC proteins (FCGBP, MUC5B and MUC1) in tissue are presented in Fig. 5. *FCGBP* is a mucus-associated IgGFc-binding protein that has been associated with a role in wound healing and maintenance of epithelial barrier function[18]. *FCGBP* shows little-to-no difference between high and low biochemical biliary viability groups (Fig. 5a). However, there is a slight trend towards higher transcript expression in baseline tissue of high histological BDI, high biochemical biliary viability livers (Fig. 5a). Marginal positive correlations between *FCGBP* raw count from baseline tissue, and protein intensity in the bile can be observed (Fig. 5b). In addition to baseline transcripts, we also saw a significant correlation of transcripts vs protein abundance in *MUC5B*; a gel-forming mucin associated with the health of various mucosal tissues[19] (Fig. 5c, d), and *MUC1*; associated with cell signaling and proliferation pathways[20] (Fig. 5e, f). While mostly positive, statistically significant correlations of transcript raw count and protein abundance are most prominent at 30 min (Fig. S8), and more so in high BDI than low BDI livers (Fig. S9). Correlations for all top-30 AUC proteins are given in Figs. S8, S9, S10 (30 min; All livers, BDI high and low, respectively) and Figs. S11, S12, S13 (150 min; All livers, BDI high and low, respectively).

To assess the cellular origin of these target proteins on a single-cell level, we accessed a publicly available single-cell RNA sequencing atlas of human livers[21]. Transcripts of *MUC1*, *MUC5B* were present in a variety of cell types, but predominant expression was shown in cholangiocytes. *FCGBP*, while again present in a variety of cell types including hepatocytes and cholangiocytes was predominantly expressed by mononuclear phagocytic cells (Fig. S14).

## Discussion

In this large-scale clinical proteomics and transcriptomics study, we identified profound bile proteome changes reflective of early injury responses, alterations in the immune response, and cellular signaling alterations over the course of ex situ normothermic machine perfusion

of human donor livers. Notably, we observed early changes in the biliary proteome indicative of cellular regeneration of the biliary tree. Pro-regenerative changes were markedly increased in livers that demonstrated a bile composition that fulfilled predefined biliary viability acceptance criteria, which resulted in safe transplantation virtually free of ischemic cholangiopathy. Moreover, we identified three proteins strongly associated with favorable bile composition and were further able to reinforce these findings in liver tissue transcripts prior to machine perfusion.

In recent years, machine perfusion technologies have been clinically implemented as a method of dynamic preservation and functional assessment of ECD livers. Although machine perfusion technologies contribute to a more rational selection of donor livers and has the potential to increase the donor pool, cholangiopathies resulting from injury to the biliary tree persist as a significant obstacle in ECD liver transplantation.

Current NMP procedures designed for preservation and assessment are relatively short (over several hours)[5,7,8] and likely explain why a large proportion of identified proteins that change over time are related to cellular signaling and acute immune responses. A remaining question to be addressed is whether, in addition to benefits in preservation and functional assessment, the quality of sub-optimal ECD livers can be improved during ex-situ perfusion. Interestingly, we observed that significant changes in regeneration-related proteins are present in the bile as early as 30 min of NMP. These proteins include mucins and mucin-related proteins (MUC5B, MUC1, FCGBP), polycomb group protein (PCG), heat shock proteins such as Clusterin (CLU) and a variety of matrix metalloproteases (MMPs), all of which have documented roles in cell cycle progression, developmental regulation and cellular proliferation in healthy and carcinogenic tissue[18,22–24]. While there is no current evidence that short-term NMP improves the quality of the organ, the initiation of regenerative processes in early stages, as defined by the current study, may help guide interventions for long-term NMP over several days[25–27]. This could be of particular use in livers considered to have poor biliary viability. Regeneration of biliary epithelial cells in an ex-situ NMP environment isolated from systemic immune responses, prior to transplantation, could circumvent injury associated with development of ischemic cholangiopathy and improve the utilization rate of ECD livers.

It is well documented that significant loss of biliary epithelium follows ischemic cold preservation and subsequent warm reperfusion[17], which is in line with the varying degree of histological bile duct injury seen in the current study, and is further reflected in our data by an initial flush-out of cellular debris proteins in the bile. Importantly, within the time points measured, we observed evidence of decreasing cellular injury throughout the perfusion. The extent of initial biliary injury is associated with the formation of post-transplant cholangiopathy, suggesting that a critical mass of functional biliary epithelium is essential for biliary health[17,28]. Our results align with these observations, where livers with low histological biliary injury are more likely to have favorable bile chemistry and therefore acceptable for transplantation. However, our primary focus is on livers with high histological bile duct injury where the outcome of biochemical biliary viability assessment is less predictable, suggesting that histological injury on its own is insufficient, and that other influential mechanisms besides injury may be present. It has been proposed that not only the extent of injury, but the ability to regenerate the biliary epithelium drives the risk for post-transplant cholangiopathy[17,29]. Specifically, we previously demonstrated that regeneration of biliary epithelium after static cold storage originates from the deep PBGs[30]. In this context, it is relevant to note the high presence of mucins and mucin-related proteins that we identified in bile formed during NMP. As these gel-forming glycoproteins are mainly secreted by the peribiliary glands, they may therefore reflect viability of these critical components of the bile duct wall[31]. Furthermore, various mucins (in particular MUC1),

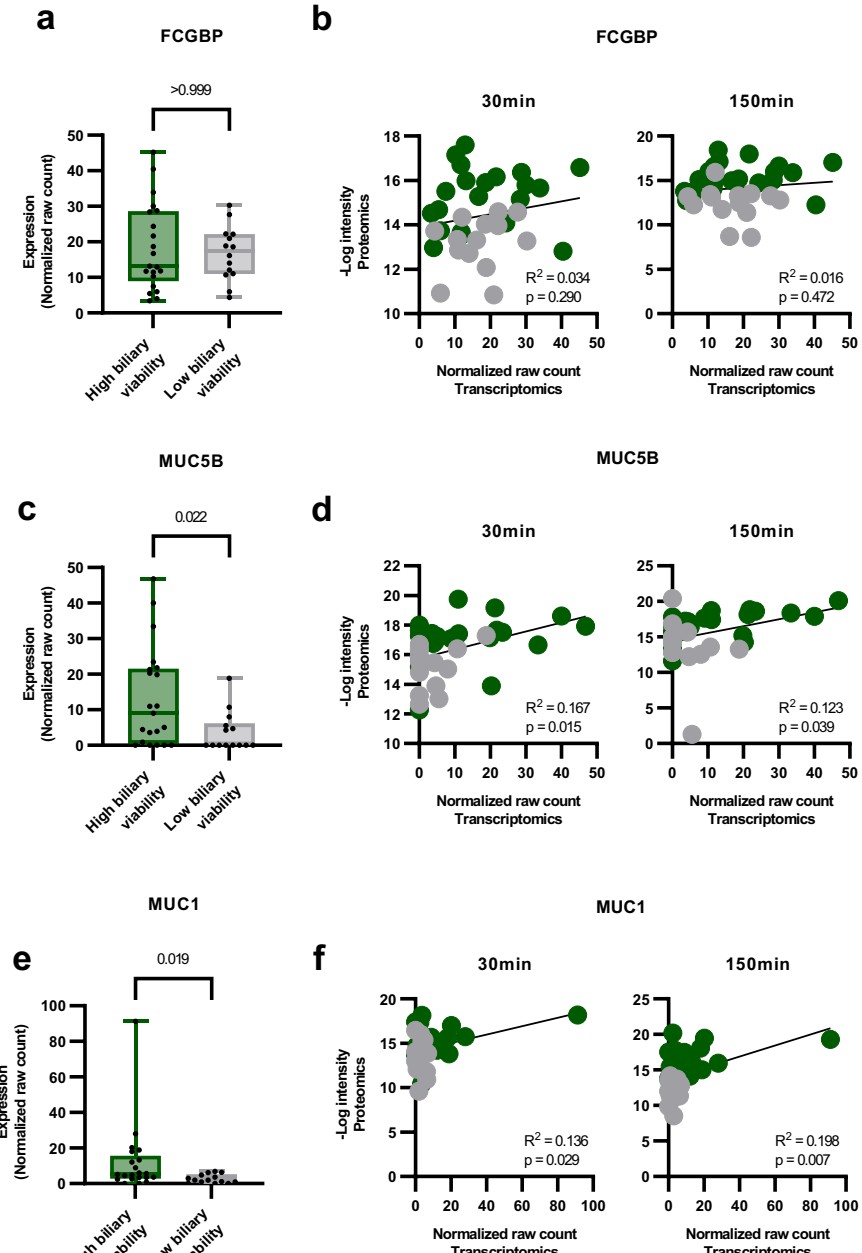

**Fig. 5 | Transcriptomic analysis of liver tissue biopsies prior to machine perfusion. a** Box plots showing individual normalized raw count transcript expression for FCGBP in high ($n = 21$) and low ($n = 14$) BDI livers and (**b**) correlation of transcript raw counts (x-axis) with protein intensity (-Log10; y-axis) within high and low BDI groups at 30 min and 150 min time points ($n = 35$). $R^2$ and correlation $p$ value are displayed for each comparison. Livers with high biliary viability score are represented by green, low biliary viability score with gray. The same graphs are depicted for MUC5B (**c**, **d**) and MUC1 (**e**, **f**). Data in box plots are presented as mean values showing 25th to 75th percentiles, with whiskers for minimum and maximum values. Box plot $p$ values were calculated using two-tailed Mann–Whitney test. Correlation $p$ values were performed using simple linear regression.

were significantly downregulated in rat livers damaged by extended cold ischemia times and were associated with increased injury of the large bile ducts following transplantation. Moreover, the formation of NAS predominantly occurs in the larger bile ducts, correlating with the abundance of mucins. This suggests that MUC1 may serve as a potential biomarker for assessing biliary injury and pathogenesis of NAS[32].

The biliary proteome of livers with favorable bile chemistry rapidly adopts a regenerative phenotype. These events suggest that biliary viability may be more determined by early regenerative and functional responses than to the extent of initial injury of the biliary epithelium. To this regard, we observed increased abundances of regenerative proteins in livers with high biochemical biliary viability scores that were acceptable for transplantation, regardless of the severity of histological bile duct injury. Maintenance and restoration of the biliary epithelium post-injury appear to be key mechanisms in the production of favorable, good quality bile. Interestingly, regulation of regenerative responses through protein-coding genes, such as *PCG*, may be crucial in differentiating between uncontrolled carcinogenic growth and regulated cellular proliferation[23].

The process of sequential hypothermic oxygenated machine perfusion, rewarming and NMP may be an initial trigger of regenerative

processes. Hypothermic oxygenated machine perfusion has been shown to restore cellular adenosine triphosphate and mitigate the deleterious effects of ischemia reperfusion injury, potentially protecting cells with regenerative capacity from extensive ischemic injury[33,34]. While we cannot rule out initiation of regenerative processes in the cold, the reduced metabolic rate and absence of bile production make this option unlikely and, importantly, difficult to address experimentally. The initiation of regenerative processes more likely occurs in the controlled rewarming phase, when cellular metabolism increases with increasing temperatures and the production of bile begins. Unfortunately, the earliest bile samples are contaminated with preservation solution that was used for bile duct flushing at the donor center, and is unlikely to accurately reflect 'real' bile composition. Therefore, we have used bile collected after 30 min of NMP as the earliest time point for our bile analyses. We cannot rule out that regenerative processes may have already been initiated during, for example, the end of the rewarming phase.

While livers with favorable bile composition show regenerative profiles, the abundance of proteins involved in the GO pathway of small molecule metabolic processes in livers with inadequate bile chemistry suggests persistent metabolic and translational (dys)regulation. Specifically, upregulation of eukaryote initiation factors (EIFs), implicated in ribosomal complex formation necessary for protein synthesis, suggest compensatory mechanisms as a suboptimal cellular repair response to biliary injury[35]. Furthermore, dehydrogenases associated with metabolic processes, such as NDUFA11, ALDH161A and fatty acid transport proteins such as CPT2 may signify key disruptions in regulation of cellular metabolism, although specific functions of these proteins in the liver have not been well established[36].

Based on our current work, we have identified potential targets for clinical application as a diagnostic test for assessing biliary viability in donor livers, including three proteins that are significantly upregulated in bile collected during NMP from livers that gave favorable bile chemistry and went on to successful transplantation. These 3 proteins (FCGBP, MUC5B and MUC1) are all closely associated and function in mucosal epithelial tissue. Although we focused on the proteome of bile to better comprehend biliary viability and functionality, we cannot make a distinction between hepatocellular or cholangiocellular origins of the proteins. By using publicly available single-cell RNA sequencing data of human livers, however, we were able to localize and visualize gene expression of proteins of interest (including, but not restricted to *MUC1*, *MUC5B* and *FCGBP*) across multiple cell types (Fig. S14)[21]. Although previous studies have shown *FCGBP* upregulation in the context of mucosal cells that healed through enhanced proliferation and migration, single cell data demonstrates that despite *FCGBP* expression in hepatocytes and cholangiocytes, it is predominantly expressed in mononuclear phagocytes[18]. Indeed, *FCGBP* has been well-characterized in distinct regenerative and anti-inflammatory actions originating from monocytes/macrophages during NMP of human livers[37]. Moreover, these cytoprotective effects have been demonstrated specifically in mucosal tissues[38], supporting the findings of our present study. Its functional role as a contributor to the protective mucous barrier in this context within the bile (ducts), further suggests that production of mucin-related proteins is crucial in driving biliary viability during NMP. The regenerative effects of FCGBP have also been characterized in malignancies, for example gall bladder cancer, as a key regulator of TGF-1-induced epithelial mesenchymal transition, whereby epithelial cells lose their cell polarity and adhesive properties to become migratory mesenchymal stem cells[39]. Under severe acute injury, such as that experienced during warm and cold ischemia, liver progenitor cells can also contribute to repopulation of the biliary epithelium. These cells also compose liver progenitor cell niches, which include macrophages (the main origin of FCGBP in the liver), and can regulate proliferation and stem cell differentiation through cellular signaling (Fig. S14)[40].

MUC1 is a cell surface protein present on the majority of apical epithelial cells and has extracellular signaling functions. Publicly available single-cell RNA sequencing data demonstrates minimal expression of *MUC1* in most liver cell types, including hepatocytes, but *MUC1* is mostly expressed in cholangiocytes (Fig. S14)[21]. *MUC1* has an influential role in variety of cell proliferation pathways, including vascular proliferation though VEGF pathways in mucosal tissues, and metastatic carcinoma progression through JAK/STAT and phosphorylation signaling pathway interactions, leading to endothelial mesenchymal transition[20,22,41,42]. Similarly, *MUC5B* was also predominantly expressed in cholangiocytes (Fig. S14), and has been associated with protection and regeneration of mucosal epithelium in airway disorders such as fibrosis and chronic obstructive pulmonary disorder. However, *MUC5B* is relatively understudied in biliary epithelium[19,21,43].

Comparisons of high and low biochemical biliary viability scores and acceptance for transplantation were based on livers that fulfilled pre-established viability criteria[8]. Although all transplanted livers resulted in excellent clinical outcomes, we inherently do not know the outcome of non-transplanted livers and acknowledge the possibility that our acceptance criteria may be too strict. That being said, there is no current gold standard for viability criteria and the utilization rate for our protocol is approximately 65-70%[8], which is comparable to other centers[7,44], but with very low incidence of post-transplant cholangiopathy[15]. Another potential limitation of this study is that the assessed histological bile duct injury was determined from the extra-hepatic bile ducts and may not be representative of injury to the entire biliary tree. Additional injury to the extrahepatic bile duct biopsy may be incurred during collection. Furthermore, the smaller, more proximal bile ducts contribute for a large degree to biochemical bile composition[31]. These parts of the biliary tree are typically less susceptible to ischemia reperfusion injury, in contrast to the larger, more distal bile ducts. To this regard, transcriptomic analysis of a peripheral liver tissue biopsy including small bile ducts identified significant differences in target genes which correlate with those in the bile proteome, strengthening the application of bile composition as a proxy for biliary injury.

Our current analysis comprises a comprehensive resource of biliary proteome-specific changes during NMP of human donor livers. Proteomes indicate that early induction of regenerative processes in response to donor process-induced biliary injury precedes favorable biochemical biliary viability and function, and subsequent successful transplantation. Our bile proteomics and liver transcriptomics data sets provide the potential to further evaluate and refine clinical viability criteria based on compositional patterns. These data pave the way for future studies on the mechanisms of ischemia-reperfusion injury, ischemic cholangiopathy, and biliary preservation and regeneration in liver transplantation.

## Methods
### Study design
This research complies with all relevant ethical regulations. For the current study, recipient data were extracted from the TransplantLines Biobank and Cohort Study of the University Medical Center Groningen (ClinicalTrials.gov Identifier: NCT03272841). The TransplantLines study protocol was approved by the institutional review board (protocol 2014/077) of the University Medical Center Groningen. Written informed consent from all participants was previously obtained.

Donor livers undergoing clinical NMP viability assessment between 30th March 2019–30th June 2022 were used for this study. All donor livers were procured by one of the regional multi-organ procurement teams in the Netherlands, using a previously described, standardized procedure[45]. Livers that were nationally declined for regular transplantation were accepted for NMP. A mandatory, five-minute no-touch period after circulatory arrest was observed for all

DCD donors. Following procurement, livers were transported to our center using static cold storage. Upon arrival, the livers were prepared for machine perfusion as previously described[8]. Donor characteristics, including agonal phase, cold/warm ischemia, and hepatectomy times are presented in Table 1.

NMP were carried out as previously described[46], using a Liver Assist device (XVIVO, Groningen, the Netherlands). To minimize reperfusion injury at the start of NMP, livers first underwent at least one hour of DHOPE (10 °C) using University of Wisconsin (UW) Machine Perfusion Solution (PumpProtect, Carnamedica, Warsaw, Poland), as described previously[34]. Following a perfusate switch to a red blood cell based perfusate, livers were rewarmed (20–37 °C) with oxygen over one hour, before maintaining NMP (37 °C) for 150 min until viability assessment. Samples were collected during all perfusion stages at predefined time points. Bile collection began at the start of NMP, when the liver started to produce bile. Bile samples for proteomics were taken at 30 min after start of NMP, at viability assessment after 150 min NMP and, for transplanted livers only, at a common time point towards the end of NMP (End). Additionally, extrahepatic bile duct and parenchymal biopsies were taken prior to machine perfusion while the liver was prepared on ice at the back table (Fig. 1a).

### Viability assessment
Hepatocellular and biliary viability were assessed at 150 min of NMP. The viability criteria used are outlined in Table S2, and published previously[8,47]. Blood gas measurements from circulating perfusate and bile were performed point-of-care, using a blood-gas analyzer (ABL 90 Flex blood gas meter, Radiometer, Brønhøj, Denmark). Based on these criteria, the decision to transplant or not transplant the organ was made at 150 min of NMP. If a liver did not meet the defined criteria, it was declined for transplantation and the perfusion terminated. If a liver met the criteria and was deemed suitable for transplantation, the liver remained on the perfusion device and NMP continued during the recipient hepatectomy. When the hepatectomy was finished, the liver was removed from the perfusion device, flushed with 2 l of cold UW preservation solution, and transported to the recipient operating room. Implantation was performed using the piggyback or classical implantation technique.

### Histology
Distal bile duct biopsies from livers undergoing NMP were taken prior to start of DHOPE. All biopsies were formalin-fixed, paraffin-embedded and cut into 4 μm sections. Sections were stained with Hematoxylin and Eosin (H&E) stain for morphological assessment. Stained sections were assessed independently by three individuals for scores in four areas: mucosal loss, vascular lesions and luminal and deep peri-biliary gland injury, as described previously[16]. The scores were then compared, any discrepancies discussed and rectified as a group.

### Proteomics
One-hundred microliters of crude bile was centrifuged at $15,000 \times g$ for 15 min to remove debris. Protein concentrations were determined using a micro-BCA protein assay kit (Cat #23235, Thermo Scientific), and were digested using in-gel digestion. Briefly, 50 μg of protein was combined with 6.25 μl 4x Loading Buffer (Abcam) and topped up to 25 μl using 100 mM ammonium bicarbonate (ABC). Samples were heated to 80 °C for 5 min, cooled, added to a gel (4–12% Bis-Tris plus, Thermo Scientific) and run for 5 minutes at 100 V. The gels were then stained for 1 h using Coomassie Blue stain (InstantBlue, Abcam), and destained in MilliQ water overnight. The protein containing gel bands were then cut from the gel and further destained/dehydrated using steps of 30% acetonitrile (ACN) in 100 mM ABC, 50% ACN in 100 mM ABC, and 100% ACN before being dried. The proteins were reduced and alkylated using 10 mM DTT at 57 °C for 30 min, and 55 mM IAA in the dark at room temperature for 30 min, respectively. The gel pieces were then dehydrated using 100% ACN and dried. The proteins were then digested using sequencing-grade trypsin (Cat #V5111, Promega) at a trypsin-to-protein ratio of 1:100 and left to digest at 37 °C overnight. The digested peptides were extracted from the gel pieces using 5% formic acid, 75% ACN. Solid-phase extraction C18 columns (S-Pure, Waters) were used for sample clean-up. The extracted peptides were then dried in a vacuum concentrator (Concentrator plus, Eppendorf), after which peptides were resuspended in 50 μl 0.1% formic acid ready for analysis by mass spectrometry (MS). An overview of the proteomics workflow can be seen in Fig. 1b.

Mass spectrometric analyses were performed on a quadrupole orbitrap mass spectrometer equipped with a nano-electrospray ion source (Orbitrap Exploris 480, Thermo Scientific). Chromatographic separation of the peptides was performed by liquid chromatography (LC) on a nano-HPLC system (Ultimate 3000, Dionex) using a nano-LC column (Acclaim PepMapC100 C18, 75 μm x 50 cm, 2 μm, 100 Å, Dionex, buffer A: 0.1% v/v formic acid in milliQ-H2O, buffer B: 0.1% v/v formic acid in acetonitrile). Six microliters of trypsin-digested bile sample were injected from a cooled autosampler (5 °C), using buffer A as a transport liquid, and loaded onto a trap column (μPrecolumn cartridge, Acclaim PepMap100 C18, 5 μm, 100 Å, 300 μm x 5 mm, Dionex). Peptides were separated on the nano-LC column using a linear gradient from 2-80% buffer B over 120 mins at a flow rate of 300nL/min. The mass spectrometer was operated in positive ion mode and data-independent acquisition mode (DIA) using isolation windows of 12 m/z with a precursor mass range of 300–1200. The orbitrap resolution was set to 120,000 and DIA scan resolution at 30,000.

### Preparation, sequencing, and analysis of RNA-seq libraries
RNA was isolated using Qiagen RNeasy Lipid Tissue Mini Kit, according to the manufacturers protocol. The RNA-seq libraries were constructed according to the Smart-3SEQ protocol[48] using 100 ng of total RNA per sample as input. The libraries were pooled together and sequenced on an Illumina NextSeq 500 instrument at ERIBA Research Sequencing Facility (Groningen, the Netherlands). Reads we trimmed using TrimGalore v.0.6.7, deduplicated with SeqKit v.2.4.0[49] and mapped to human genome assembly GRCh38.p14 using Hisat2 v.2.2.1[50]. Gene counts were generated by featureCounts v2.0.1[51] using NCBI RefSeq gene annotations. Normalization of the mapped read counts was performed using the DEseq2 software package in R[52].

### Data analysis and statistics
MS raw data was searched using Spectronaut software (version 14, Biognosys) using library-free data independent acquisition (DIA) analysis workflows. Oxidation and deamidation were set as variable modificaitons and carbamidomethylation as a fixed modification. Data was searched against human protein sequences using the UPR_homoSapiens_30-09-2021 FASTA file (20,386 entries). Identified DIA protein group quantitation values were processed using Perseus software (v1.6.6.4). The data were log2 transformed and missing values replaced using imputation based on normal distribution of values. Batch correction was performed using the R plug-in in Perseus, using the COMBAT method. Volcano plots were made using parametric Students T-tests (assuming normal distribution of data) with a permutation-based FDR of 0.05 to assess multiple comparisons. Heat maps were generated using R after log2 transformed values were converted to z-score, with hierarchical clustering of proteins and samples.

Biological pathway and functional enrichment analysis was performed in CytoScape 3.9.1 (National Resource for Network Biology) software using STRING protein databases (https://string-db.org/). Visualization of volcano plots, principal component analyses, correlation analysis and box plots were made using GraphPad Prism 8 (San Diego, CA, USA) software. Receiver operator characteristic (ROC)

curve and binary logistic regression analyses was performed using SPSS version 28 (IBM) software.

## Reporting summary

Further information on research design is available in the Nature Portfolio Reporting Summary linked to this article.

## Data availability

Raw mass spectrometry data used for proteomic analysis are available through ProteomeXchange via PRIDE, with the identifier PXD046355. RNA sequencing data used for transcriptomics analysis are available through the European Nucleotide Archive with the identifier E-MTAB-13501. All other data are available in the main text or supplementary materials. Single-cell RNA sequencing data was used from a publicly available dataset (http://liveratlas-vilarinholab.med.yale.edu), using data deposited at the NCBI GEO (accession numbers: GSE115469, GSE136103, GSE129933, GSE124395 and GSE130473)[21]. Source data are provided with this paper.

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

## Acknowledgements

V.M. is supported by a VENI research grant by the Dutch Research Council (NWO; grant #09150161810030). V.M. further reports a Research grant from the Dutch Ministry of Economic Affairs (Health~Holland Public Private Partnership grant #PPP31 2019-024), and a Research grant from the Dutch Society for Gastroenterology (NVGE #01-2021), both outside the submitted work. F.K. is supported by an unrestricted grant from the Noaber Foundation, Lunteren, The Netherlands.

## Author contributions

A.T. performed the research, analysed the data and wrote the paper. J.W., K.U., E.B., analysed the data and reviewed the manuscript.B.L., S.B., V.L., O.L., and I.J. performed the research and reviewed the manuscript.T.L. and F.K. participated in the research design and reviewed the manuscript.R.P. participated in the research design, performed the research, and reviewed the manuscript.V.M. participated in the research design, performed the research, analysed the data and wrote the paper.

## Competing interests

The authors declare no competing interests.

## Additional information

**Peer review information** : *Nature Communications* thanks Dagmar Kollmann and the other, anonymous, reviewer(s) for their contribution to the peer review of this work. A peer review file is available.

