## [Peer Review File · Nature Communications]

Bile proteome reveals biliary regeneration during normothermic preservation of human donor liversEditorial Note: Parts of this Peer Review File have been redacted as indicated to remove third-party material where no permission to publish could be obtained.

REVIEWER COMMENTS

Reviewer #1 (Remarks to the Author):

This original article focuses on the assessment of bile composition during NMP. Bile obtained during NMP has been subjected to proteomic analyses and significant differences have been found between livers with a high vs. low biliary viability score. Several proteins have been identified that have been associated with regenerative processes. The research presented is novel and the proteomic analysis of bile during NMP is an important step to further understand the mechanisms of ischemia-reperfusion injury and to obtain better tools to evaluate biliary injury during NMP. The 55 livers (35 of them transplanted) that have been included in the study are well defined and thoroughly investigated. The article is well written and clear.

There are a few points that should be considered prior to publishing:

- 1) It is not clear from the abstract and the title, that the livers have been subjected to DHOPE and COR prior to NMP. This should be clarified.
- 2) DHOPE and COR itself have several effects on biliary injury and the development of cholangiopathy. Livers might start regenerative processes earlier than if they would have been subjected to NMP right after static cold storage. Furthermore, several proteins might have been flushed out during DHOPE and COR. Please comment on the effects of DHOPE and COR in the discussion and how this might have affected the proteomic analysis during NMP.
- 3) It would be interesting, to understand the differences between proteomics performed in bile vs. perfusate. Do the authors have data on proteomics from the perfusate? Is it possible to test for the predominant proteins identified in bile (FCGBP, MUC5B and MUC1) in perfusate and compare between the livers? This might help to better understand if the described processes are bile specific.
- 4) Furthermore, the identified proteins should be evaluated in perfusate obtained during

DHOPE and COR.

5) Typos:

line 107: ..., 33 had bile samples available at (instead of and)

line 408: ..., when the (delete of the) liver started to produce bile

Reviewer #2 (Remarks to the Author):

Groningen's group has been working on liver recovery from extended criteria donors with NMP for long. Herein, Adam M. Thorne et al presents a well-described study on bile proteomic analysis during NMP of human donor livers. Generally, the manuscript is well-written with splendid data, while the results based on the bioinformation analysis are still weak to support the conclusion.

Some issues in the current study are still worth further discussion.

1. Groningen's group published the research on biomarkers of biliary injury and restoration, including bile production, biliary bicarbonate, pH, bicarbonate, glucose gamma-glutamyltransferase, lactate dehydrogenase and bile duct biopsy et al (Iris E M de Jong et al. Transplantation, 2023 Jun 1;107(6):e161-e172; Sanna Op den Dries, et al. Liver Transpl, 2016 Jul;22(7):994-1005). No evidence showed the significant correlation between bile chemistry and bile duct histology during 150 min NMP. The current study used only bile duct histology to define high/low biliary viability.

Bile parameters are necessary to be supplemented to distinguish biliary viability.

2. Compared with a recently published article on scRNA in liver NMP in Nature Communication (T Hautz, et al. Nat Commun, 2023 Apr 21;14(1):2285), expression and functional assessment of the selected genes/proteins (including FCGBP, MUC5B and MUC1) from bile proteomics and liver transcriptomics in the current human study is missing and short of the necessary validation.

Thus, the further verification of protein expression and function should be supplemented.

The preclinical NMP model could be helpful.

RESPONSE TO REFEREES

Bile proteome reveals biliary regeneration during normothermic preservation of human donor livers

Reviewer #1:

This original article focuses on the assessment of bile composition during NMP. Bile obtained during NMP has been subjected to proteomic analyses and significant differences have been found between livers with a high vs. low biliary viability score. Several proteins have been identified that have been associated with regenerative processes. The research presented is novel and the proteomic analysis of bile during NMP is an important step to further understand the mechanisms of ischemia-reperfusion injury and to obtain better tools to evaluate biliary injury during NMP. The 55 livers (35 of them transplanted) that have been included in the study are well defined and thoroughly investigated. The article is well written and clear.

There are a few points that should be considered prior to publishing:

1. It is not clear from the abstract and the title, that the livers have been subjected to DHOPE and COR prior to NMP. This should be clarified.

We appreciate the reviewer's positive feedback and their recognition of the novelty and importance of our study. The reviewer is correct in that we have performed sequential dual hypothermic oxygenated machine perfusion (DHOPE), controlled oxygenated rewarming (COR), followed by normothermic machine perfusion (NMP), as specified in the methods section (lines 450-455). However, the main focus of our manuscript is about bile proteome and good quality bile is only produced by the liver at normothermia (i.e., during NMP). In addition, because of the Nature Communications authors guidelines which stipulate a maximum of 15 words in the title (now 12), we felt that adding additional words to the title would distract from the main message. To accommodate the reviewer, we have further clarified our protocol also in the abstract of the revised manuscript (lines 32-34) with the following text:

“Fifty-five human donor livers underwent sequential hypothermic machine perfusion, rewarming and NMP. We performed an in-depth, unbiased proteomics analysis of bile collected during NMP.”

2. DHOPE and COR itself have several effects on biliary injury and the development of cholangiopathy. Livers might start regenerative processes earlier than if they would have been subjected to NMP right after static cold storage. Furthermore, several proteins might have been flushed out during DHOPE and COR. Please comment on the effects of DHOPE and COR in the discussion and how this might have affected the proteomic analysis during NMP.

We acknowledge that DHOPE is associated with a reduced incidence of post-transplant cholangiopathy in the recipient by mitigation of ischemia-reperfusion injury, which was previously

published by our group (Van Rijn, et al. *New Engl J Med* 2021)¹ and others (Schlegel, et al. *EBioMedicine* 2020)². In the cold, the metabolic rate of the liver is approximately 10-15% (please see the graph below).

[Editorial Note: figure redacted]

Figure: Graphic presentation of the change in the rate of metabolism with decreasing temperature. Metabolic rate during DHOPE (8-12°C; listed as hypothermic machine perfusion [HMP] on this graph) is approximately 10-15%. Metabolic rate is based on Van't Hoff's principle (expressed as $Q_{10} = (k_2/k_1)^{10/(t_2-t_1)}$). From: Karangwa, et al. *Am J Transplant* 2016.³

Because of the low metabolic rate, bile production is absent during hypothermia, and therefore makes it impossible to collect and analyze during cold perfusion. Although we cannot rule out the presence of regenerative processes of the biliary tract in the cold, absence of bile production prohibits biliary proteome analyses. At the end of the rewarming phase, the first bile production starts typically at temperatures above 30°C. This first bile is often heavily diluted with UW solution due to flushing of the bile ducts in the donor hospital, which we feel does not accurately reflect bile composition or the condition of the bile ducts. For this reason, we chose 30 minutes into NMP for our first bile sample time point in this study, where all livers are producing bile. We agree with the reviewer that it is certainly possible that regenerative processes begin as the liver warms during COR and metabolic rate rises, although it remains problematic to obtain samples that adequately and accurately reflect the phenotype at this time point. We have specifically addressed this in the revised manuscript (discussion; **lines 351-363**) with the following text:

“The process of sequential hypothermic oxygenated machine perfusion, rewarming and NMP may be an influencing factor in the early abundance of regenerative processes. Hypothermic oxygenated machine perfusion has been shown as a resuscitative method to restore cellular adenosine triphosphate and mitigate the deleterious effects of ischemia reperfusion injury, potentially protecting cells with regenerative capacity from extensive ischemic injury (33, 34). While we cannot currently rule out initiation of regenerative processes in the cold, the reduced metabolic rate and absence of bile production make it both unlikely and difficult to measure. The initiation of such processes is more probable in the following controlled rewarming phase, when cell metabolism increases with increasing temperatures and production of bile begins. Unfortunately, these earliest bile samples mostly comprise of preservation solution that was used for bile duct flushing at the donor center, and is unlikely to accurately reflect bile composition. Therefore, the earliest time point for our bile analyses was after 30 min of NMP. We cannot rule out that regenerative processes may have already been initiated during, for example, the end of the rewarming phase.”

3. It would be interesting, to understand the differences between proteomics performed in bile vs. perfusate. Do the authors have data on proteomics from the perfusate? Is it possible to test for the predominant proteins identified in bile (FCGBP, MUC5B and

¹ van Rijn, R., et al., Hypothermic Machine Perfusion in Liver Transplantation - A Randomized Trial. *N Engl J Med*, 2021. 384(15): p. 1391-1401.

² Schlegel, A., et al., Hypothermic oxygenated perfusion protects from mitochondrial injury before liver transplantation. *EBioMedicine*, 2020. 60: p. 103014.

³ Karangwa, S.A., et al., Machine Perfusion of Donor Livers for Transplantation: A Proposal for Standardized Nomenclature and Reporting Guidelines. *Am J Transplant*, 2016. 16(10): p. 2932-2942.

MUC1) in perfusate and compare between the livers? This might help to better understand if the described processes are bile specific.

During the design of our study, we hypothesized that the composition of bile (and not of the perfusate) would most likely reflect the condition of the biliary tree. It is an interesting suggestion by the reviewer to investigate the three proteins (i.e., FCGBP, MUC5b and MUC1) that we found in bile, also in the perfusate. We have therefore analyzed protein abundance of these three proteins during 30min, 150min and End of NMP in the perfusate. As in the present study, livers were stratified by time point and by high or low biochemical biliary viability score. Interestingly, MUC1 was not detectable by mass spectrometry in the perfusate at any of the time points. Please see details of MUC5B and FCGBP in the graphs below.

These data show that the abundance of MUC5b and FCGBP are not different between groups with a high or low biliary viability score over time, and that machine perfusion perfusate is therefore less likely to reflect the condition of the biliary tree. We did, however, perform transcriptomic analysis of the abovementioned three proteins in liver tissue, which includes hepatocellular bile producing cells as well as small bile ducts (Figures S5, S6, S7). Here, we identified significant differences in genes relating to the top-30 AUC proteins (high vs low biliary viability) found in the bile. These analyses strengthen the notion that bile composition (and not perfusate) is a valid proxy for biliary injury (Figures S8, S9, S10 and S11, S12, S13).

4. Furthermore, the identified proteins should be evaluated in perfusate obtained during DHOPE and COR.

Between DHOPE and COR, we apply a switch of perfusion solution from an acellular preservation solution into a red-blood-cell-based perfusion solution. Perfusate samples during HOPE, therefore, cannot be compared to those during COR of NMP and were thus not collected. In addition, due to the low metabolic rate of the liver at hypothermic temperatures (see also our response to question 2), gene transcription and protein abundance are expected to be very low. Although we acknowledge that an evaluation of the abundance of these proteins during the COR (rewarming) phase would potentially be of interest, given the underwhelming results even during NMP (with a fully metabolically active liver) as presented above (response to question 3), we speculate that protein abundance during COR would not

provide additional information and feel that these analyses are beyond the scope of the current study.

5. Typos:

line 107: ..., 33 had bile samples available at (instead of and)

line 408: ..., when the (delete of the) liver started to produce bile

*The typos have been corrected in the revised manuscript (**line 110 and 456**).*

Reviewer #2:

Groningen's group has been working on liver recovery from extended criteria donors with NMP for long. Herein, Adam M. Thorne et al presents a well-described study on bile proteomic analysis during NMP of human donor livers. Generally, the manuscript is well-written with splendid data, while the results based on the bioinformation analysis are still weak to support the conclusion.

Some issues in the current study are still worth further discussion:

1. Groningen's group published the research on biomarkers of biliary injury and restoration, including bile production, biliary bicarbonate, pH, bicarbonate, glucose gamma-glutamyltransferase, lactate dehydrogenase and bile duct biopsy et al (Iris E M de Jong et al. Transplantation, 2023 Jun 1;107(6):e161-e172; Sanna Op den Dries, et al. Liver Transpl, 2016 Jul;22(7):994-1005). No evidence showed the significant correlation between bile chemistry and bile duct histology during 150 min NMP. The current study used only bile duct histology to define high/low biliary viability. Bile parameters are necessary to be supplemented to distinguish biliary viability.

*We thank the reviewer for the critical appraisal of our manuscript and appreciate the valuable comments provided. We would like to clarify the assessment of biliary viability. We fully agree that functional, point-of-care biochemical bile parameters are currently essential to assess cholangiocellular viability, like we have published previously as the reviewers correctly points out. We would like to confirm that the decision to transplant the livers in our study was based on pre-defined, biochemical viability criteria (**lines 95-97**; and **Table S2**, below; full functional parameters of each liver at the moment of viability assessment are provided in **Table S3**). Histological analysis of bile duct injury was performed retrospectively, and did not influence the decision to transplant. Livers that showed low levels of histological bile duct injury were more likely to have favorable biochemical biliary viability scores (78% vs 22%, **Figure 2C**). However, high histological injury was unable to discern between high and low biochemical biliary viability scores. It is possible that increased histological injury may be an artifact of the biopsy collection process, and may not be representative of actual injury. We feel this further confirms that histological injury assessed using the extrahepatic bile duct is insufficient to independently inform biliary viability. Biochemical cholangiocellular viability, and subsequent proteomics analysis, may support a better reflection of biliary health and function. We apologize that this distinction was not made clearer, and have further clarified in the results (**line 123**) and discussion section (**lines 327-328 and lines 423-424**).*

Table S2. Viability criteria for determining liver viability at 150mins NMP.

	Parameter	Green zone	Orange zone	Red zone
Hepatocytes	Bile production (mL)	≥ 10*	5 to 10	<5
	Perfusate lactate (mmol/L)	< 1.7	1.7 to 4.0	> 4.0
	Perfusate pH	7.35 – 7.45	7.25 to 7.35	< 7.25
Cholangiocytes	Bile pH	> 7.45	7.40 to 7.45	< 7.40
	ΔpH	> 0.10	0.05 to 0.10	< 0.05
	ΔHCO ₃ ⁻ (mmol/L)	> 5.0	3.0 to 5.0	< 3.0
	ΔGlucose (mmol/L)	< -5.0	-3.0 to -5.0	> -3.0
Score		2	1	0

Viability criteria that needed to be reached within 150min of normothermic machine perfusion. The green zone includes the four original viability criteria (perfusate pH, lactate, bile production and bile pH) that had to be reached at any time point within 150min after initiation of NMP. The other criteria were secondary criteria that emerged with increasing experience. Orange zone represents potentially acceptable values which are 'on the border', and that could be accepted if the other viability criteria are 'green'. Red zone indicates values that do not meet the viability criteria.

** Of which ≥ 4 mL in the last hour. Δ indicates the bile value minus the perfusate value. Abbreviations: NMP; normothermic machine perfusion.^{4,5}*

2. Compared with a recently published article on scRNA in liver NMP in Nature Communication (T Hautz, et al. Nat Commun, 2023 Apr 21;14(1):2285), expression and functional assessment of the selected genes/proteins (including FCGBP, MUC5B and MUC1) from bile proteomics and liver transcriptomics in the current human study is missing and short of the necessary validation. Thus, the further verification of protein expression and function should be supplemented. The preclinical NMP model could be helpful.

We thank the reviewer for their thoughtful comments regarding the characterization and function of the mentioned proteins. We performed in-depth, unbiased proteomics using data independent acquisition, followed by bulk RNA sequencing, to identify proteins in the bile throughout the perfusion and transcripts in liver tissue at the earliest possible time point we have available. We further describe the potential role and function of these mentioned proteins in the context of the present study. We feel that the addition of a less specific, lower resolution assay, such as Western blot, would not provide additional value to the present

⁴ van Leeuwen, O.B., et al., Sequential hypothermic and normothermic machine perfusion enables safe transplantation of high-risk donor livers. Am J Transplant, 2022. 22(6): p. 1658-1670.

⁵ O. B. van Leeuwen, Y. de Vries, V. E. de Meijer, R. J. Porte, Hypothermic machine perfusion before viability testing of previously discarded human livers. Nat Commun 12, 1008 (2021).

study. Further to this, we would like to bring to the reviewer's attention a recently published correspondence in a Nature journal highlighting the incongruency of such assays in validation of proteomics data.⁶ Common drawbacks of Western blot are antibody non-specificity and semi-quantitative interpretation. Therefore, we thank the reviewer for highlighting the recent, important article by Hautz. et al published in Nature Communications, which provides an in-depth characterization of immune cells present in liver normothermic machine perfusion. To better characterize our findings at a higher resolution as the reviewer suggests, we accessed a publicly available single-cell RNA sequencing atlas of 28 human livers.⁷ Using this platform, we searched the three identified proteins (MUC1, MUC5B, and FCGBP) for RNA expression in the single-cell context of human liver (we kindly direct the attention of the reviewer to the image below).

Figure: Single-cell RNA sequencing data of MUC1, MUC5B and FCGBP in healthy human livers. Visualization of gene expression across multiple cell types (localization in top panel, in purple; expression level in bottom panel). MUC1 and MUC5B are expressed in a variety of cell types, including hepatocytes. However, the highest level of expression is in cholangiocytes. While FCGBP is also expressed in cholangiocytes, its predominant expression is in mononuclear phagocytes. All three described proteins are involved in the initiation and

⁶ Mehta, D., et al., The incongruity of validating quantitative proteomics using western blots. Nat Plants, 2022. 8(12): p. 1320-1321.

⁷ Brancale, J. and S. Vilarinho, A single cell gene expression atlas of 28 human livers. J Hepatol, 2021. 75(1): p. 219-220.

regulation of regenerative processes within the biliary epithelium, including endothelial mesenchymal transition. Data displayed was obtained from a publicly available RNA sequencing data set.⁷

In the single-cell RNA sequencing atlas, MUC1 and MUC5B were minimally present in a variety of cell types, including hepatocytes, but their gene expression was mostly present in cholangiocytes (**see figure**). MUC1 is the best characterized mucin. It is a transmembrane protein that has multiple effects on transcription factors, mitochondrial proteins and, similar to FCGBP, affects endothelial to mesenchymal transition. Furthermore, MUC1 regulates proliferative processes, including JAK/STAT signaling, Wnt signaling and VEGF pathways, amongst others.⁸ While less extensively characterized than MUC1, MUC5B is one of the major gel-forming mucins produced. Its role relates to protection of the endothelium through mucus formation, immune response regulation and initiation of regeneration.

FCGBP is expressed, to some extent, by cholangiocytes and hepatocytes, but its gene expression is mainly present in mononuclear phagocytes. In the paper highlighted by the reviewer (Hautz et al. Nat Commun 2023), the authors demonstrated a distinct regenerative and anti-inflammatory action of monocytes/macrophages during normothermic perfusion of human livers, supporting our findings and the suggested origins of FCGBP in the context of the present study. FCGBP has been described as having cytoprotective and anti-inflammatory roles, specifically in mucosal tissues.⁹ Its functional role as a contributor to the protective mucous barrier, in this context within the bile (ducts), further suggests that production of mucin-related genes/proteins are crucial in driving biliary viability in the context of NMP. Furthermore, FCGBP has been well characterized, for example in gallbladder cancer, as a key regulator of TGF-1-induced epithelial-mesenchymal transition, whereby the epithelial cells lose their cell polarity and adhesive properties to become migratory mesenchymal stem cells.¹⁰ We suggest in the context of the present study, that the presence of secreted FCGBP we find in bile with favorable biochemistry may indicate initiation of regenerative processes to repopulate the biliary epithelium. To further interpret the role of FCGBP, it has been described that under severe acute injury, such as that experienced during warm and cold ischemia, liver progenitor cells can also contribute to repopulation of the biliary epithelium. These can also compose liver progenitor cell niches, which include macrophages (the main origin of FCGBP in the liver), and can regulate proliferation and stem cell differentiation through cellular signaling.¹¹

With regard to the reviewer's excellent suggestion, in the revised manuscript we have added additional information to the discussion (**lines 379-409**) and have added the figure (shown above) to the supplementary material (**Supplementary Figure 14**) to better interpret the roles of these key proteins in the context of the present study.

⁸ Kasprzak, A. and A. Adamek, Mucins: the Old, the New and the Promising Factors in Hepatobiliary Carcinogenesis. Int J Mol Sci, 2019. 20(6).

⁹ Liu, Q., et al., Role of the mucin-like glycoprotein FCGBP in mucosal immunity and cancer. Front Immunol, 2022. 13: p. 863317.

¹⁰ Xiong, L., et al., NT5E and FcGBP as key regulators of TGF-1-induced epithelial-mesenchymal transition (EMT) are associated with tumor progression and survival of patients with gallbladder cancer. Cell Tissue Res, 2014. 355(2): p. 365-74.

¹¹ Lan, T., et al., Role of Immune Cells in Biliary Repair. Front Immunol, 2022. 13: p. 866040.

REVIEWERS' COMMENTS

Reviewer #1 (Remarks to the Author):

The authors have adequately responded to the questions raised.

Reviewer #2 (Remarks to the Author):

Generally, the manuscript has been improved after the careful modification.

Firstly, I am fine with the response and modification based on the first issue.

Regarding to the second issue, more efforts are needed.

1. Please add scRNA analysis and result into the manuscript based on the scRNA analysis of the public database.
2. The title describes the proteomics analysis of bile, while the actual manuscript comprises proteomics, RNA sequence and scRNA . Thus, RNA sequence and scRNA should be reflected in the title too, like multi-omics analysis of...
3. The whole analysis is based on the retrospective study. Please emphasize the significance of the selected genes/proteins as biomarker or therapeutic target. Expression and functional assessment of the selected genes/proteins should be added, as well as the correlation between the expressions of the selected targets and the clinical data. The preclinical NMP model could be helpful if the correlation between the selected genes/proteins and clinical data is weak.

RESPONSE TO REFEREES (2)

Bile proteome reveals biliary regeneration during normothermic preservation of human donor livers

Reviewer #2:

1. Please add scRNA analysis and result into the manuscript based on the scRNA analysis of the public database.

We thank the reviewer for the critical appraisal of our revised manuscript and appreciate the additional comments. We have already included the scRNA analysis data in the manuscript (Figure S14) and provided detailed explanations in the results (lines 276-280) and discussion sections (lines 381-387, 397-399, and 401-408). We believe this adequately addresses the reviewer's request.

2. The title describes the proteomics analysis of bile, while the actual manuscript comprises proteomics, RNA sequence and scRNA. Thus, RNA sequence and scRNA should be reflected in the title too, like multi-omics analysis of..

While the primary focus is on extensive bile proteomics, RNA sequencing and scRNA analysis were conducted on liver tissue to support our findings in bile. Because the majority of data concerns the proteome of bile collected during normothermic machine preservation, we believe our current title, "Bile Proteome Reveals Biliary Regeneration During Normothermic Preservation of Human Donor Livers" accurately represents the primary focus.

3. The whole analysis is based on the retrospective study. Please emphasize the significance of the selected genes/proteins as biomarker or therapeutic target. Expression and functional assessment of the selected genes/proteins should be added, as well as the correlation between the expressions of the selected targets and the clinical data. The preclinical NMP model could be helpful if the correlation between the selected genes/proteins and clinical data is weak.

In our previous rebuttal letter, we've addressed the reviewer's question regarding "further verification of protein expression and function" by including additional data on scRNA (as mentioned above, in both the results and discussion as well as in Figure S14). In Figure S8 and Figure S11, we already provide correlations of RNA sequencing transcripts from baseline biopsies with bile protein intensity for 30 different proteins at 30min and 150 min, respectively. In Table S4, we have provided individual area under the curve values for the same top 30 highest AUC proteins identified in livers with high biliary viability. Of note, average AUC of the top-3 proteins (FCGBP, MUC5b and MUC1) was 0.90, 0.86 and 0.85, respectively. We believe that conducting additional work beyond this would extend beyond the scope of the current study.